# Benefits of Adhering to a Mediterranean Diet Supplemented with Extra Virgin Olive Oil and Pistachios in Pregnancy on the Health of Offspring at 2 Years of Age. Results of the San Carlos Gestational Diabetes Mellitus Prevention Study

**DOI:** 10.3390/jcm9051454

**Published:** 2020-05-13

**Authors:** Verónica Melero, Carla Assaf-Balut, Nuria García de la Torre, Inés Jiménez, Elena Bordiú, Laura del Valle, Johanna Valerio, Cristina Familiar, Alejandra Durán, Isabelle Runkle, María Paz de Miguel, Carmen Montañez, Ana Barabash, Martín Cuesta, Miguel A. Herraiz, Nuria Izquierdo, Miguel A. Rubio, Alfonso L. Calle-Pascual

**Affiliations:** 1Endocrinology and Nutrition Department, Hospital Clínico Universitario San Carlos and Instituto de Investigación Sanitaria del Hospital Clínico San Carlos (IdISSC), 28040 Madrid, Spain; veronica.meleroalvarez10@gmail.com (V.M.); carlaassafbalut90@hotmail.co.uk (C.A.-B.); nurialobo@hotmail.com (N.G.d.l.T.); i.jimenez.varas@gmail.com (I.J.); elena.bordiu@salud.madrid.org (E.B.); lauradel_valle@hotmail.com (L.d.V.); valeriojohanna@gmail.com (J.V.); cristinafamiliarcasado@gmail.com (C.F.); aduranrh@hotmail.com (A.D.); irunkledelavega@gmail.com (I.R.); pazdemiguel@telefonica.net (M.P.d.M.); mcmnita@hotmail.com (C.M.); ana.barabash@gmail.com (A.B.); cuestamartintutor@gmail.com (M.C.); marubioh@gmail.com (M.A.R.); 2Facultad de Medicina. Medicina II Department, Universidad Complutense de Madrid, 28040 Madrid, Spain; maherraizm@gmail.com (M.A.H.); nuriaizquierdo4@gmail.com (N.I.); 3Centro de Investigación Biomédica en Red de Diabetes y Enfermedades Metabólicas Asociadas (CIBERDEM), 28029 Madrid, Spain; 4Gynecology and Obstetrics Department, Hospital Clínico Universitario San Carlos and Instituto de Investigación Sanitaria del Hospital Clínico San Carlos (IdISSC), 28040 Madrid, Spain

**Keywords:** glucose tolerance, Mediterranean diet, nutritional intervention, obesity, offspring, pregnancy nutrition

## Abstract

The intrauterine environment may be related to the future development of chronic diseases in the offspring. The St. Carlos gestational diabetes mellitus (GDM) prevention study, is a randomized controlled trial that evaluated the influence of the early (before 12th gestational week) Mediterranean diet (MedDiet) on the onset of GDM and adverse gestational outcomes. Out of 874 women assessed after delivery (440 control group (CG)/434 intervention group (IG)), 703 children were followed (365/338; CG/IG), with the aim to assess whether the adherence to a MedDiet during pregnancy induces health benefits for the offspring during the first two years of life. Logistic regression analysis showed that the IG in children of mothers with pre-gestational body mass index (BMI) < 25 kg/m^2^ and normal glucose tolerance (NGT), was associated with a lower risk (RR(95% CI)) of suffering from severe events requiring hospitalization due to bronchiolitis/asthma (0.75(0.58–0.98) and 0.77(0.59–0.99), respectively) or other diseases that required either antibiotic (0.80(0.65–0.98) and 0.80(0.65–0.99), respectively), corticosteroid treatment (0.73(0.59–0.90) and 0.79(0.62–1.00) respectively) or both (all *p* < 0.05). A nutritional intervention based on the MedDiet during pregnancy is associated with a reduction in offspring’s hospital admissions, especially in women with pre-gestational BMI < 25 kg/m^2^ and NGT.

## 1. Introduction

Adherence to healthy eating patterns in pregnancy, such as the Mediterranean diet (MedDiet), is being widely studied. There seems to be associations between the development of certain diseases that can affect pregnancy, such as gestational diabetes mellitus (GDM), and the onset of immune and metabolic diseases in the offspring later in life [1,2,3,4,5,6,7,8,9,10,11,12,13,14,15,16,17,18,19,20,21].

Evidence suggests a possible beneficial effect of following appropriate eating habits during pregnancy in terms of disease development. Indeed, the MedDiet seems to have a protective role against diseases such as bronchiolitis and also in diseases of an autoimmune nature such as asthma, wheezing, allergic rhinitis, atopic dermatitis and food allergies in childhood [1,2,13,15,16,17,18]. Conversely, a western diet can have the opposite effects [3,19,20]. Furthermore, recent findings point out that pregnancy could be an optimal time to establish suitable eating patterns in the mother, thus guaranteeing the offspring’s health [4].

The adverse intrauterine environment provided by either GDM or obesity is linked to epigenetic changes that predispose the offspring to develop a metabolic disease later in life. In turn, these can be transmitted to the following generation, thereby perpetuating the vicious cycle of metabolic diseases [5,6,7,8,9,10]. In fact, several studies have shown how maternal obesity and GDM during pregnancy are associated with an increase in the risk of asthma in early childhood. This has been observed even in women without a history of asthma [11]. Moreover, having GDM and obesity are associated with a higher risk for the offspring of developing respiratory diseases and of having poorer health [12,14,21].

Recently, it has been shown that adherence to a MedDiet can reduce the risk of GDM [22]. In fact, our group has shown that an early adherence to a MedDiet‒supplemented with extra virgin olive oil (EVOO) and nuts‒in pregnancy can reduce the risk of GDM and other adverse materno-fetal outcomes [23,24]. It has also been associated with a better postpartum metabolic profile in the mother [25]. Whether these benefits are conveyed to the offspring, remains to be known.

While the current evidence suggests a possible association between diet and the development of diseases in children, few have evaluated the effect of an intervention based on a Mediterranean diet in the development of metabolic and immune diseases in the offspring. In addition, the results of these studies are heterogeneous and have not been developed as a randomized controlled trial (RCT), so they are not conclusive.

This study aims to assess whether the MedDiet supplemented with EVOO and pistachios during pregnancy induces benefits to the offspring’s health during the first two years of life.

## 2. Materials and Methods

### 2.1. Study Design

This is a prospective analysis of the St. Carlos GDM prevention study [23]. This paper includes offspring of women who attended the postpartum follow-up between 2017 and 2018.

Concisely, this RCT evaluated whether an early nutritional intervention based on a MedDiet (supplemented with EVOO and pistachios) could reduce the incidence of GDM. Women in the intervention group (IG) were told to enhance the consumption of EVOO and pistachios while those in the control group (CG) were told to restrict all kinds of fats.

After the delivery, women received the same dietary recommendation.

The study was approved by the Ethics Committee of Hospital Clínico San Carlos (full protocol approved 17 July 2013 (CI 13/296-E)) and conducted according to the Helsinki Declaration. All women signed a letter of informed consent.

This trial was registered on 4 December 2013, at http://www.isrctn.com/ with the number ISRCTN84389045 (DOI 10.1186/ISRCTN84389045). The authors confirm that all ongoing and related trials for this intervention are registered.

### 2.2. Study Population

To obtain children’s data, the 874 women who were analyzed in the St. Carlos GDM prevention study were invited to participate in the follow-up at 2 years postpartum.

A total of 171 (75 from CG and 96 from IG) did not attend the 2-year follow-up. Of these, it was not possible to access the children’s data because of changes in the place of residence to outside of the community of Madrid, making it impossible to contact the mothers and to access their children’s medical records. Thus, the total of children assessed was 703 (80.5%), 365 from CG and 338 from IG (Figure 1).

### 2.3. Outcomes

Primary outcome: to assess the incidence of bronchiolitis/asthma, atopic dermatitis and food allergies as well as the number and duration of all-cause hospital admissions in children at 2 years of age. Secondary outcomes: to evaluate the rates of hospital admissions due to severe episodes of bronchiolitis/asthma and other diseases requiring pharmacological treatment with antibiotics, corticosteroids or both.

### 2.4. Data Collection

The hospital Clínico San Carlos is a hospital within the public health system that provides health care services for a population of about 380,000 habitants of the central area of Madrid. The public health system covers health care at pediatric age, including medical and nursing consultations, and provides free access to the mandatory vaccination program. It also facilitates access to optional vaccines. At the first visit with the pediatrician and nurse after birth, usually before the first month, the child receives a health card. This card includes data about their anthropometric development, the vaccines received and the introduction of different foods. The medication prescribed by the pediatrician is recorded in the electronic history and dispensed in the pharmacy after being included in the unique prescription module (MUP). When hospital admission is required, the diagnosis and treatment received are recorded in the discharge report and can be accessed through the HORUS program (Historia Clínica Digital del Sistema Nacional de Salud).

#### 2.4.1. Clinical Data: Mothers

The maternal data referred to in this study belong to mothers whose children were evaluated in this study. These data were obtained during their gestational period.

This information included family history of metabolic disorders such as type 2 diabetes, obstetric history (miscarriages and GDM), educational level, employment status, number of prior pregnancies, smoking habits (registering whether they are currently smoking, or they smoked until they found out they were pregnant) and gestational age at entry concurring to the first ultrasound.

The mother’s lifestyle during and after pregnancy was registered. The adherence to a healthy lifestyle (including physical activity) was evaluated with the Diabetes Nutrition and Complication Trial (DNCT) questionnaire and provided the nutrition score and physical activity score. The adherence to the MedDiet was assessed with the 14-point Mediterranean diet Adherence Screener (MEDAS) questionnaire and was used to obtain the MEDAS score. A more detailed description has been previously published [23].

#### 2.4.2. Clinical Data: Children

Children’s data were obtained at 2 years of age after a face to face interview with the mother. The interview was conducted by a dietician. It was carried out in the hospital and its duration was about 30 min. They brought the mandatory pediatric health registry (primary source) to this visit. Anthropometric data (weight and height taken at different time points), vaccination schedule (compulsory and optional vaccination) and food introduction during the first two years were obtained from this mandatory pediatric health registry. Other data about the children’s health had to be obtained from a secondary data source. If the mothers did not attend this visit, these data were recovered by contacting them via telephone calls. All information provided by the mother was later verified with the data found in the secondary sources.

Secondary data source: Information about prescription of pharmacological treatments, including antibiotics and corticosteroids and number of episodes requiring pharmacological treatment, was obtained from MUP. Information about the number of hospital admissions and their cause was obtained from the hospital discharge registry. The number of minor diseases (requiring only outpatient treatment) and their treatment were obtained from the electronic medical history system (through the HORUS application) and SERMAS (Servicio Madrileño de Salud) where all information related to children’s health is registered. This includes attendance to emergency room, hospitalization (cause, duration and discharge are registered), diseases diagnoses, allergies and the pharmacological treatment received.

Moreover, the vaccination schedule can also be accessed through the electronic medical history, and includes the moment of reception and the dose of the vaccine received. All this information could be retrieved as long as the child was attended to within the Community of Madrid. Access to the electronic medical history system outside of this area was not possible.

The following variables were extracted: (1) Food allergy or intolerance, bronchiolitis/asthma and atopic eczema. It was recorded at the time of diagnosis along with the number of episodes in which their pediatrician prescribed antibiotic treatment, corticosteroids or both. (2) Vaccination schedule: this includes the mandatory vaccines in Spain (chickenpox, diphtheria, hemophilus influenza type B, hepatitis B, measles, meningococcus C, mumps, pertussis, pneumococcus 13 V, polio, rubella and tetanus) and the optional ones (meningococcus B and rotavirus). It was considered complete when they received all the recommended doses. (3). The total number of all-cause hospital admissions and their durations. Additionally, the number of hospitalizations due to severe episodes of bronchiolitis/asthma, as well as the number and duration of hospital admissions in which the children required antibiotic or corticosteroids treatment, or both. This information had to be recorded in the hospital discharge report and was accessed through the HORUS program.

The following additional data were also registered: (1) Breastfeeding: recorded as either exclusive lactation; mixed, which includes any product (artificial lactation) or complementary feeding; registering the moment the children started consuming cereals (both with and without gluten). (2) Kindergarten attendance: whether they attended and the age they started (measured in months).

### 2.5. Statistical Analysis

The categorical variables are expressed numerically (%) and continuous variables are expressed as mean (SD). The comparison of frequencies between groups of the categorical variable was evaluated by the χ^2^ test. For continuous variables, values were compared with Student’s t test or the Mann–Whitney U test if distribution of continuous variables was not normal, as verified by the Shapiro–Wilk test. Logistic regression analysis was used to assess the adjusted effect of the intervention on the risk for adverse offspring outcomes that were significantly different in the binary analysis. The magnitude of association was evaluated using the relative risk (RR) and 95% confidence interval (CI) adjusted for age, ethnicity and parity and categorized by BMI and glucose tolerance, were estimated. The reason for adjusting age, ethnicity and parity was because advanced maternal age, parity and ethnicity are associated with worse health in the offspring [26,27,28,29]

All *p* values are 2-tailed at less than 0.05. Analyses were performed using SPSS, version 21 (SPSS, Chicago, IL, USA).

## 3. Results

A total of 703/874 (80.5%) children of women who completed the St. Carlos GDM prevention study were evaluated: 365 of the CG and 338 of the IG. Compared to women in the CG, women of the IG were similar in relation to the baseline characteristics at 12 weeks of gestation (GW). However, they maintained a greater adherence to the nutritional intervention (as reflected by the MEDAS and nutrition score), lower rates of GDM, lower fasting glucose levels and less weight gain at 24–28 GW. The mothers’ baseline characteristics are shown in Table 1.

A non-significant reduction in the rates of all-cause hospital admissions and in the number of children who required it was observed in the IG versus the CG. In total, there were 51 (15.1%) and 65 (17.8%) hospital admissions, respectively, and 46 (13.6%) and 54 (14.9%) children hospitalized respectively, both *p* > 0.05. However, the length of stay was significantly shorter in IG than in the CG (6.8 ± 9.1 vs. 11.9 ± 25.2 days; *p* = 0.02). Data of the children of mothers from the CG and IG at 23 ± 2.5 months of age are shown in Table 2

The Appendix A show the data of the children according to the maternal pre-pregnancy body mass index (BMI) and by glucose tolerance, comparing the IG versus the CG. When evaluating children born to women with pre-gestational BMI < 25 kg/m^2^, a significantly lower hospital admissions rate due to bronchiolitis/asthma, antibiotic treatment and corticosteroid treatment, was observed in the IG (all *p* < 0.05) (Appendix A). There were no significant results found when comparing the children of mothers with BMI ≥ 25 kg/m^2^. Similar results were observed when analyzing the children of women who had normal glucose tolerance (NGT), where a significantly lower number of hospital admissions due to bronchiolitis/asthma, antibiotic treatment and corticosteroid treatment was also observed in the IG (all *p* < 0.05). These results were not found when making these same comparisons between children of mothers who developed GDM (Appendix A).

When analyzing children of mothers with pre-gestational BMI < 25 kg/m^2^ of the IG, the RR (95% CI) in the IG of having a severe event requiring hospital admission due to bronchiolitis/asthma, due to any disease requiring antibiotic treatment, either any disease requiring corticosteroid treatment or both, were 0.75 (0.58–0.98), 0.80 (0.65–0.98), 0.73 (0.59–0.90) and 0.78 (0.61–0.99), respectively. In children of mothers with NGT, the RRs were 0.77 (0.59–0.99), 0.80 (0.65–0.99), 0.75 (0.60–0.93) and 0.79 (0.62–1.00), respectively, for children of mothers with NGT, respectively. These data are shown in Figure 2.

## 4. Discussion

This study shows that the adherence to the MedDiet enriched with EVOO and pistachios during pregnancy seems to be associated with a lower risk of hospitalization in children at two years of age. This was especially observed in women who had a pre-gestational BMI < 25 kg/m^2^ and in those with NGT. To our knowledge, this is the first RCT study that analyses the influence of a nutritional intervention based on a MedDiet in pregnancy on the offspring’s health.

The MedDiet enriched with EVOO and pistachios has been associated with health benefits in the mother [25,30,31]. There are also suggestions that these benefits can be transferred to postnatal life [32]. In fact, recent published studies seem to indicate that the MedDiet decreases the incidence of wheezing, asthma and allergies [1,2,18]. A confounding factor could be the caesarean section rates since it has been shown that the immune system of children born by caesarean section matures later and therefore they have a higher risk of developing allergies in the future [33]. The results showed similar data in both groups. Thus, a reduction in infectious and allergic events would be expected. However, we have not found differences between the IG and CG in relation to rates of bronchiolitis/asthma, diseases of autoimmune origin (food allergies and dermatitis eczema) and infectious illness that did not require hospital admission. Adherence to childhood vaccination programs is associated with a reduction of these diseases during the first years of life [34]. Considering this, the vaccination program was completed by more than 99% of children in both groups. That is to say, it could overcome the benefit of food during pregnancy. On the other hand, it is known that breastfeeding is the greatest protective factor in offspring during the early years of life, preventing early childhood diseases and several infectious diseases [35,36]. The present study revealed that more than 90% of women (from both the IG and CG) performed exclusive breastfeeding at least during the first five months of life. In many cases, mothers continued combining breastfeeding with complementary feeding during at least 10 months. This could be the reason why we have not found more significant differences between children at two years of age. In fact, some studies have found certain differences in older children [37]. Therefore, it would be interesting to evaluate these children at an older age.

Nevertheless, our study shows significant differences between the IG and CG in relation to the reduction of severe events requiring hospitalization in children whose mothers had a pre-gestational BMI < 25 kg/m^2^ and NGT. These are considered low-risk women, who make up most of our studied population.

Indeed, according to the results, the rates of corticosteroid treatment in children would be higher in the IG, but these are not statistically significant. On the other hand, the reduction in the hospitalized rate due to severe adverse effects is highly significant.

GDM and obesity have negative effects on the offspring, and when both coexist these effects are enhanced [7,8,10,38]. In our study, no significant differences were observed between women from the IG and CG in those with a BMI ≥ 25 kg/m^2^ or with GDM. This could suggest that the nutritional intervention used in our study may not be strong enough to decrease the incidence of severe events in children, of high-risk mothers, at this age. A longer follow-up period may clarify whether these benefits can be observed in those born to high-risk mothers.

EVOO and pistachios, rich in phenolic components, are associated with a better anti-inflammatory, immunomodulatory and microbiota profile. In pregnancy, changes occur in the mother’s immune system, affecting normal gut function and composition of the microbiota. These changes, which could also be affected by the mother’s diet, could enhance the long-term health of the mother and her offspring [39,40,41,42].

Several limitations were found in our study. First, differences in diet between groups may have not been wide enough to induce changes in offspring health. Both groups (IG and CG) received recommendations based on the Mediterranean diet, reinforced or restricted in the consumption of fats. The differences obtained in the score of the questionnaires used between both groups were over two points during pregnancy, a difference that can be considered insufficient to detect statistically significant differences.

Secondly, a major limitation is that some data had to be completed through the register. This does not allow all data to be extracted in the same way. However, these data were always recorded by the same person, thus avoiding differences in data collection; therefore, it is unlikely that this could affect our results.

Lastly, breastfeeding, vaccination and other relevant factors within the first two years of life could have influenced the results. For instance, the rate of SGA and LGA newborns, although it is significantly lower in the IG, the number is small enough to affect our results. However, its effect in a wider cohort, or an older age, cannot be ruled out. Therefore, a follow-up at two-years of age may be insufficient to detect statistically significant differences in non-serious diseases at this age. Consequently, a study evaluating children at 5–6 years of age is ongoing.

Our results strengthen the recommendation to implement the MedDiet‒reinforced with EVOO and pistachios‒during pregnancy, since it also seems to provide health benefits for the offspring, at least in those born to low-risk women. Whether these results are sustained over the time, remains to be known.

## 5. Conclusions

A nutritional intervention based on the MedDiet during pregnancy seems to be associated with a reduction in the offspring’s hospital admissions requiring antibiotic and corticosteroid treatment, and admissions related to asthma/bronchiolitis, especially in women who have pre-gestational BMI <25 kg/m^2^ and NGT.

## Figures and Tables

**Figure 1 jcm-09-01454-f001:**
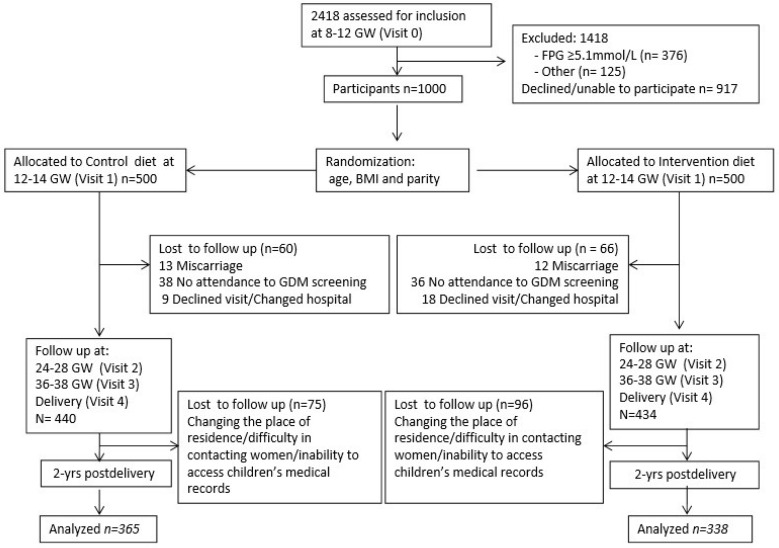
The CONSORT 2010 flow diagram for scope.

**Figure 2 jcm-09-01454-f002:**
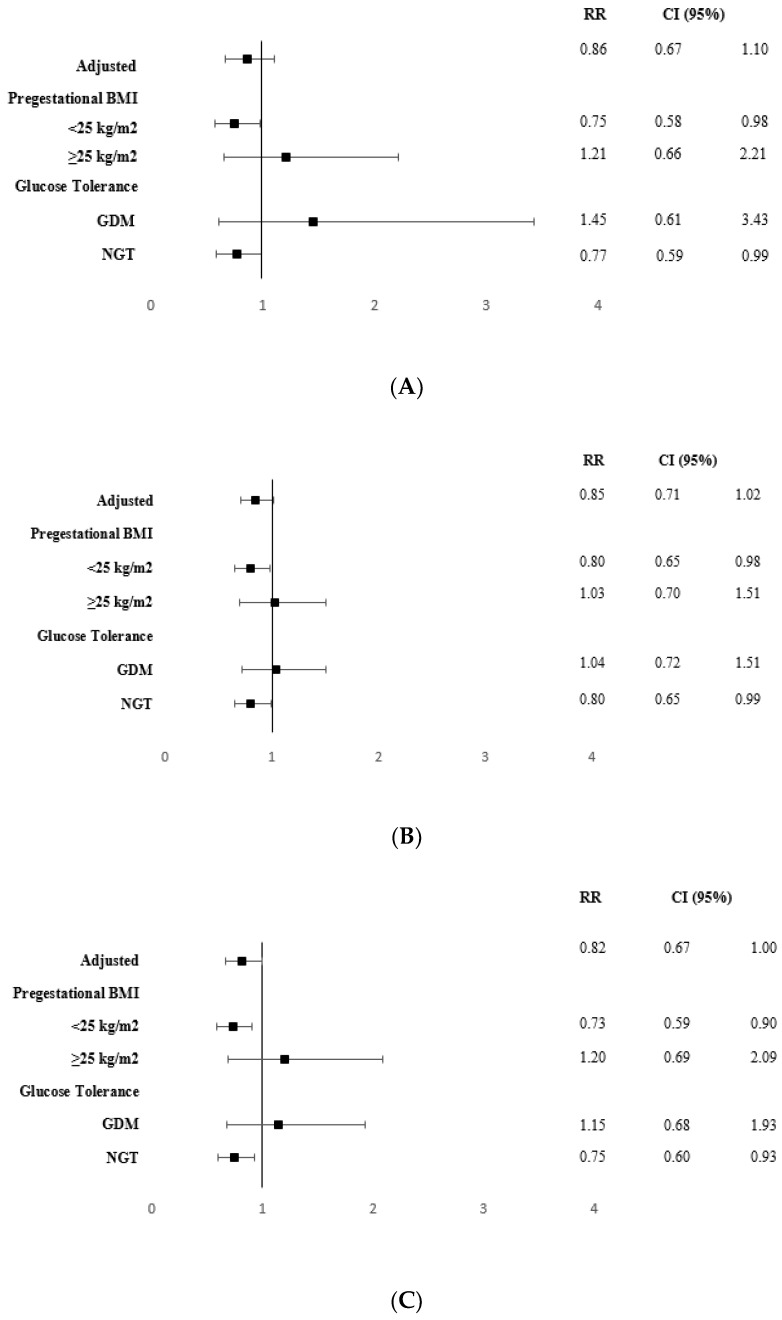
Relative risk (RR) and 95% confidence interval (CI) of suffering severe adverse events that require hospitalization adjusted for age, parity and ethnicity, and categorized by BMI and glucose tolerance. (**A**) Bronchiolitis/asthma in hospitalized children; (**B**) antibiotic treatment in hospitalized children; (**C**) corticosteroid treatment in hospitalized children; (**D**) antibiotic and corticosteroid treatment in hospitalized children. BMI, body mass index; RR, relative risk; CI, confidence interval; GDM, gestational diabetes mellitus; NGT, normal glucose tolerance. Regression analysis was assessed with SPSS, version 21 (reference group = control group).

**Table 1 jcm-09-01454-t001:** Baseline characteristics of mothers whose children were analyzed.

	CONTROL GROUP *n* = 365	INTERVENTION GROUP *n* = 338	*p*
Age (years)	32.8 ± 5.3	33.2 ± 5.0	0.316
Race/Ethnicity			
Caucasian	237 (64.9)	222 (65.6)	
Hispanic	114 (31.2)	109 (32.2)	
Others	14 (3.9)	7 (2.2)	0.804
Family history of			
Type 2 Diabetes	82 (23.5)	92 (27.7)	
MetS (>2 components)	65 (17.8)	76 (22.5)	0.154
Previous history of			
GDM	11 (3.0)	11 (3.3)	
Miscarriages	117 (32.0)	117 (34.6)	0.688
Educational status			
Elementary education	41 (11.2)	21 (6.2)	
Secondary School	145 (39.7)	146 (43.2)	
University Degree	174 (47.7)	168 (49.7)	
UNK	5 (1.4)	3 (0.9)	0.467
Employment	276 (75.6)	266 (78.7)	0.950
Number of pregnancies			
Primiparous	144 (39.5)	142 (42.1)	
Second pregnancy	119 (32.6)	115 (34.1)	
>2 pregnancies	102 (27.9)	81 (22.8)	0.622
Smoker			
Never	202 (55.3)	184 (54.1)	
Current	26 (7.1)	26 (7.7)	0.994
Gestational Age (weeks) at baseline	12.1 ± 0.6	12.0 ± 0.3	0.899
Pre-pregnancy Body Weight (kg)	61.5 ± 11.1	60.3 ± 9.9	0.131
Weight gain at:			
24–28 GW	7.73 ± 4.22	7.04 ± 3.71	0.022
36–38 GW	11.02 ± 6.71	11.49 ± 6.87	0.452
Pre-pregnancy BMI (kg/m^2^)	23.3 ± 3.9	23.1 ± 3.5	0.354
BMI ≥ 25 kg/m^2^	107 (29.3)	86 (25.4)	0.486
Systolic BP/Diastolic BP (mm Hg)			
12 GW	107 ± 11/64 ± 9	107 ± 10/66 ± 9	0.957/0.061
24 GW	105 ± 11/63 ± 9	105 ± 11/63 ± 8	0.370/0.747
36 GW	113 ± 13/72 ± 9	113 ± 13/73 ± 9	0.319/0.303
Fasting Blood Glucose (mg/dl)			
12 GW	81.4 ± 6.1	81.2 ± 6.0	0.687
24 GW	85.8 ± 6.7	84.0 ± 6.5	0.001
36 GW	77.1 ± 7.7	75.0 ± 7.7	0.007
GDM at 24–28 GW *n* (%)	91 (24.9)	58 (17.2)	0.036
Caesarean Section *n* (%)	56 (15.3)	54 (16.0)	0.111
TSH mcUI/mL			
12 GW	1.9 ± 1.2	2.1 ± 1.4	0.223
24 GW	2.0 ± 1.2	2.1 ± 1.1	0.464
36 GW	1.7 ± 1.3	1.6 ± 0.9	0.890
MEDAS Score			
12 GW	4.1 ± 1.7	4.4 ± 1.6	0.090
24 GW	4.5 ± 1.7	6.3 ± 1.7	0.001
36 GW	5.5 ± 1.9	6.6 ± 2.1	0.001
Nutrition Score			
12 GW	0.6 ± 3.3	0.2 ± 3.1	0.078
24 GW	1.2 ± 3.4	4.2 ± 3.2	0.001
36 GW	3.6 ± 3.7	5.3 ± 3.6	0.001
Physical Activity Score			
12 GW	−1.7 ± 1.0	−1.9 ± 1.0	0.059
24 GW	−1.8 ± 0.9	−1.8 ± 0.9	0.482
36 GW	−1.8 ± 0.7	−1.6 ± 0.9	0.055

Data are mean ± SD or number (%). MetS, metabolic syndrome; UNK, unknown; BMI, body mass index; GW, gestational weeks; GDM, gestational diabetes mellitus; BP, blood pressure; MEDAS Score, 14-point Mediterranean diet Adherence Screener (MEDAS); nutrition score, Diabetes Nutrition and Complications Trial (DNCT); physical activity score, (walking daily (>5 days/week) Score 0: At least 30 min. Score+1, if >60 min. Score−1, if <30 min. Climbing stairs (floors⁄day, >5 days a week): Score 0, between 4 and 16; Score+1, >16; Score−1: <4). *p* differences between groups analyzed with the χ^2^ test (categorical variable); Student’s t test (continuous variables) or the Mann–Whitney U test (not-normal distribution in continuous variables). Verified by the Shapiro–Wilk test.

**Table 2 jcm-09-01454-t002:** Children’s data at 2 years follow-up according to whether their mothers belonged to the intervention (IG) or control group (CG).

	CG	IG	*p*
Number (*n*)	365	338	
Born, *n* (%)			
Preterm (<37 GW)	14 (3.8)	5 (1.5)	0.477
Small for gestational age (SGA)	23 (6.3)	5 (1.5)	0.002
Large for gestational age (LGA)	10 (2.7)	4 (1.2)	0.049
Age (months)	23.13 ± 2.55	23.29 ± 2.51	0.433
Body Weight (kg)	12.11 ± 1.48	12.17 ± 1.54	0.555
Percentile	47.7 ± 27.0	49.2 ± 27.4	0.483
Height (cm)	86.26 ±3.96	86.16 ± 4.01	0.759
Percentile	39.3 ± 27.0	39.7 ± 28.6	0.867
Breastfeeding *n* (%)	340 (94.4%)	314 (93.5%)	0.703
Exclusive (months)	5.20 ± 1.50	5.36 ± 1.47	0.194
Mixed (months)	10.34 ± 7.74	10.60 ± 7.55	0.705
Cereal Introduction (months)			
Gluten-free cereal	4.77 ± 0.81	4.74 ± 0.81	0.702
Gluten cereal	6.58 ± 1.28	6.62 ± 2.02	0.787
Nursery			
*n* (%)	247 (67.7)	237 (70.7)	0.213
Age (months)	15.8 ± 6.0	14.8 ± 6.8	0.139
Vaccinations *n* (%)			
Compulsory	359 (99.4%)	338 (100%)	0.270
Recommended *n* (%)			
Meningitis	210 (58.2)	214 (64.3)	0.059
Rotavirus	251 (69.5)	240 (72.1)	0.257
Outpatient diseases *n* (%)			
Treatment with antibiotics	251 (68.8%)	234 (69.4%)	0.456
Treatment with corticosteroids	187 (51.2%)	166 (49.3%)	0.327
Diagnoses *n* (%)			
Food allergies	29 (8.0%)	21 (6.2%)	0.225
Asthma	7 (1.9%)	11 (3.3%)	0.189
Bronchiolitis/respiratory problems	74 (20.3%)	75 (22.3%)	0.298
Atopic dermatitis	101 (27.7%)	106 (31.5%)	0.161
Severe Diseases inpatients treatment			
All-cause hospital stays *n* (%)	65 (17.8%)	51 (15.1%)	0.193
Children *n* (%)	54 (14.9%)	46 (13.6%)	0.079
Duration (days)	11.9 ± 25.2	6.8 ± 9.1	0.020
Asthma/bronchiolitis disease	27 (7.4%)	18 (5.3%)	0.167
Treatment with antibiotics	59 (16.2%)	40 (11.9%)	0.063
Treatment with corticosteroids	41 (11.2%)	25 (7.4%)	0.054
Treatment with antibiotics and corticosteroids	36 (9.9%)	25 (7.4%)	0.155

Results expressed as mean ± SD or *n* (%). *p* differences between groups analyzed with the χ^2^ test (categorical variable); Student’s t test (continuous variables) or the Mann–Whitney U test (not-normal distribution in continuous variables). Verified by the Shapiro–Wilk test.

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
