# Peer review of "Benefits of Adhering to a Mediterranean Diet Supplemented with Extra Virgin Olive Oil and Pistachios in Pregnancy on the Health of Offspring at 2 Years of Age. Results of the San Carlos Gestational Diabetes Mellitus Prevention Study."

_jcm, 2020, doi:10.3390/jcm9051454_

Round 1
Reviewer 1 Report
The paper by Melero et al describes the role of the Mediterranean diet with EVOO and pistachios on hospital admission in 2 years old children born from mothers trained to this dietary pattern with respect to another one focused on a low-fat diet. They showed that the Mediterranean diet, a pre-pregnancy normal-weight, and normal glucose tolerance during pregnancy seem to protect babies from hospitalization, asthma/bronchiolitis, and treatments with antibiotics/corticosteroids.
The reading of the paper suggests some comments:
- Page 7. It is unclear if only one hospital was involved. They wrote, “Information about the number of hospital admissions and their cause was obtained from the hospital discharge registry”. However, if only one hospital was included, the Authors cannot exclude that data are incomplete being the cohort so numerous. Mothers and children could be referred to other hospitals closer to their home, meanwhile for pregnancy mothers very frequently prefer the referent hospital.
- They showed the time of introduction of gluten, as well as the time of breastfeeding, etc. they collected these data from both telephone calls and paediatric health registry. However, the strength of results depends on the method of collection and ascertainment. The Authors should clarify how many results are retrieved by registries and how many by self-reports. Furthermore, this aspect should be discussed in limitations.
- The Authors presented the weight and height of children. Please, insert also SDS or Z score that give to readers a more precise idea of growth.
- Methods and results. The authors observed the importance of weight in the pre-pregnancy period. Did they control for the increase in Kgs during pregnancy? Did this point have an influence?
- Methods and results. Did the Authors correct outcomes at 2 years for some parameters at birth that could be critical as weight categories (SGA, AGA, LGA), time of birth (preterm, term), complications at birth (respiratory distress, antibiotic treatments, others)?
- The Authors reported the number of food allergies. However, food allergies at 2 years of age are difficult to diagnose and frequently are temporary and disappeared after 6 years of age. This point should be discussed.
Author Response
Thank you very much for your kind comments and constructive suggestions. We agree with you on most points. We have made the corresponding changes to improve the article.
The changes applied are as follows:
The paper by Melero et al describes the role of the Mediterranean diet with EVOO and pistachios on hospital admission in 2 years old children born from mothers trained to this dietary pattern with respect to another one focused on a low-fat diet. They showed that the Mediterranean diet, a pre-pregnancy normal-weight, and normal glucose tolerance during pregnancy seem to protect babies from hospitalization, asthma/bronchiolitis, and treatments with antibiotics/corticosteroids.
The reading of the paper suggests some comments:
Page 7. It is unclear if only one hospital was involved. They wrote, “Information about the number of hospital admissions and their cause was obtained from the hospital discharge registry”. However, if only one hospital was included, the Authors cannot exclude that data are incomplete being the cohort so numerous. Mothers and children could be referred to other hospitals closer to their home, meanwhile for pregnancy mothers very frequently prefer the referent hospital.
All women who participated in the RCT were followed-up at the Hospital Clínico San Carlos. All the required visits were performed exclusively at this hospital. However, this was irrespective to whether women gave birth at other hospitals. It was also independent to whether their children were followed-up at other hospitals or primary care centres. Despite this, we were able to retrieve most of the information from children’s medical records (HORUS program) as long as their medical centre was within the community of Madrid
In addition, we believe there must have been a misunderstanding. In the sentence paraphrased here, in Page 7 line 155, the “number of hospital admissions” does not refer to mother’s hospital admission, but to their children’s’. Everything under the section “2.4.2 Clinical data: Children” refers to children’s information.
They showed the time of introduction of gluten, as well as the time of breastfeeding, etc. they collected these data from both telephone calls and paediatric health registry. However, the strength of results depends on the method of collection and ascertainment. The Authors should clarify how many results are retrieved by registries and how many by self-reports. Furthermore, this aspect should be discussed in limitations.
These data were obtained in most part from the face-to-face interview with the mother. Or, if this was not possible, data were retrieved by contacting the mother via telephone. This information, which is shown in page 7 line 144, 150 and 151, was then verified by consulting the paediatric health registry (HORUS). This task was always done by the same person, which avoids possible confounding factors in this regard.
Following this suggestion, we have added this to the limitations section (page 16, lines 312-314).
The Authors presented the weight and height of children. Please, insert also SDS or Z score that give to readers a more precise idea of growth.
We agree on that point. Percentiles of Spanish population have just been added in the tables (page 11-12, table 2).
Methods and results. The authors observed the importance of weight in the pre-pregnancy period. Did they control for the increase in Kgs during pregnancy? Did this point have an influence?
This is an interesting comment. It would have been interesting to evaluate the effects of weight gain during pregnancy on the health of the children at 2 years of age. Unfortunately, our sample size was not enough as to perform this kind of analysis per different groups of weight gain.
Methods and results. Did the Authors correct outcomes at 2 years for some parameters at birth that could be critical as weight categories (SGA, AGA, LGA), time of birth (preterm, term), complications at birth (respiratory distress, antibiotic treatments, others)?
We only had the following rates published: GC / GI: SGA: 25/5, LGA: 18/4. Preterm < 37 GW: 17/5 R Distress 4/3 and stay in NICU 14/8. In the sample of children the rates are as follows: SGA 18/5 LGA 10/4 Preterm 14/5 Distress 3/2 and stay in NICU 10/6. Only the LGA and SGA rates are statistically different between both groups (CG vs. IG), but the rates are low in both groups, and it does not affect our results. The number of cases are insufficient to affect the analysed data. Furthermore, the antibiotic treatments received at birth were not recorded. They were only considered in our study after hospital discharge.
The Authors reported the number of food allergies. However, food allergies at 2 years of age are difficult to diagnose and frequently are temporary and disappeared after 6 years of age. This point should be discussed.
We agree with your comments. Due to the difficulty of establishing a diagnosis, we have included in one same variable both allergies and food intolerances. In page 8, line 167-168 it is mentioned the moment in which the allergy is diagnosed and registered by the paediatrician (In line 167-168 has also been added a more concise phrase). Whether they are transitory or persist in time will be considered in the follow-up at 5 years of age, as we suggest in the limitations (page 16, lines 317-318).
Reviewer 2 Report
The topic studied is a relevant topic for public health, the main strength of which is that it is the first RCT to analyze the nutritional influence based on an intervention with a Mediterranean diet in pregnancy on the health of the offspring. Although the results are not entirely positive, it may be the basis for longer-term studies. However, there are methodological aspects in this work that have not been clarified and that require more detail.
Abstract. This section requires providing more information. No relevant aspects of the trial methodology are specified and the numerical results are scarce, since, for example, a lower risk is specified, but it is so represented (RR, 95% CI). The conclusion with the results is repeated.
Introduction. The first paragraph should improve, since it includes, for example, the following meta-analysis “Mediterranean diet during pregnancy and childhood for asthma in children: A systematic review and meta-analysis of observational studies” in which it is not concluded what in that paragraph is mentioned exactly because of the high heterogeneity.
The first two sentences of the second paragraph of the introduction are repetitive, both mentioning asthma.
In general, this article lacks justification before the objective, which justifies carrying out this work more concretely. For example: Are previous studies inconsistent? Causes? What weaknesses do the other paper mentioned in the introduction have that this paper can improve?
Methodology. Specify which personnel and how many conducted the interviews with the mother in the follow-up to collect the information, as well as the characteristics of the interview: duration, location.
Results. The data shown in figure 2 refer to adjusted ORs that could be biased by other possible confounding factors. Potential confounding factors in the observed relationship should be taken into account that could, perhaps, modify the associations shown.
According to Table 2, the frequency of corticosteroid treatment for asthma, bronchiolitis, and atopic dermatitis is higher in the intervention group, although the hospitalizations are less, so it is necessary to be very cautious to justify this fact and in the conclusion.
Discussion. The first paragraph of the discussion should be expanded, since it is true that the risk of hospitalizations is lower but corticosteroid treatment is greater in the intervention group.
The reason for adjusting the results by age, ethnicity and parity is not discussed.
I am methodologically concerned with the first limitation mentioned by the authors “differences in diet between groups may have not been wide enough to induce changes in offspring health”. Since the intervention in clinical trials must be totally differentiated from the recommendations given to the control group. Furthermore, part of the second limitation could have been resolved by showing results adjusted for more potentially confusing variables.
Figures. Figure 1 does not have the title in the text. The image quality is not good and cannot be appreciated well.
Tables. It would avoid shading because it is very difficult to understand.
Author Response
Thank you very much for your kind comments and constructive suggestions. We agree with you on most points. We have made the corresponding changes to improve the article.
The changes applied are as follows:
The topic studied is a relevant topic for public health, the main strength of which is that it is the first RCT to analyze the nutritional influence based on an intervention with a Mediterranean diet in pregnancy on the health of the offspring. Although the results are not entirely positive, it may be the basis for longer-term studies. However, there are methodological aspects in this work that have not been clarified and that require more detail.
Abstract. This section requires providing more information. No relevant aspects of the trial methodology are specified and the numerical results are scarce, since, for example, a lower risk is specified, but it is so represented (RR, 95% CI). The conclusion with the results is repeated.
We agree with your comments. Information has been cut off due to the word-limitation (200 words). For this reason, we have had to reduce the information about methodology and results. We have rewritten a new abstract (with 196 words) which hopefully is more complete (pages 1-2).
Introduction. The first paragraph should improve, since it includes, for example, the following meta-analysis “Mediterranean diet during pregnancy and childhood for asthma in children: A systematic review and meta-analysis of observational studies” in which it is not concluded what in that paragraph is mentioned exactly because of the high heterogeneity.
The first two sentences of the second paragraph of the introduction are repetitive, both mentioning asthma.
We agree with your comments, maybe this is not clear enough. For this reason, these paragraphs in the introduction have been changed to adjust the text with our references and unify them. (page 3, lines 54-57 and lines 58-61).
In general, this article lacks justification before the objective, which justifies carrying out this work more concretely. For example: Are previous studies inconsistent? Causes? What weaknesses do the other paper mentioned in the introduction have that this paper can improve?
We agree. We have re-arranged the references and discussed the inconsistencies in the current scientific evidence in order to provide a justification as to why this study is important (Page 4, lines 80-84).
Methodology. Specify which personnel and how many conducted the interviews with the mother in the follow-up to collect the information, as well as the characteristics of the interview: duration, location.
We also agree on this point. The staff that carried out the interviews, as well as their location and duration have been added (page 7, lines 144-146).
Results. The data shown in figure 2 refer to adjusted ORs that could be biased by other possible confounding factors. Potential confounding factors in the observed relationship should be taken into account that could, perhaps, modify the associations shown.
The data shown in figure 2 have been adjusted for age, ethnicity and parity, as mentioned in the paper (page 9, line 193). However, in the figure we mistakenly wrote “crude” rather than “adjusted”. We have corrected this (page 13-14, figure 2).
According to Table 2, the frequency of corticosteroid treatment for asthma, bronchiolitis, and atopic dermatitis is higher in the intervention group, although the hospitalizations are less, so it is necessary to be very cautious to justify this fact and in the conclusion.
We agree with your comments and on the fact that we must be cautious in interpreting our results. However, although the rates may be higher in the intervention group, they do not reach statistical significance. Meanwhile there is a significant reduction in the rates of severe adverse events. In figure 2.a. (page 13) you can observe how in the IG women with a BMI <25 kg/m² and NGT have a lower risk of having their children being hospitalized due to bronchiolitis/asthma. This is why in the conclusion we state that the MedDiet is associated with a lower risk admissions related to asthma/bronchiolitis.
Discussion. The first paragraph of the discussion should be expanded, since it is true that the risk of hospitalizations is lower but corticosteroid treatment is greater in the intervention group.
We agree with you. For this reason, we have added a concise paragraph (page 15, lines 292-294) to clarify that the fact that, despite having obtained higher rates of corticosteroid treatment needed in minor events in the IG, the rates of hospitalization requirement in severe events with corticosteroid treatment were significantly lower.
The reason for adjusting the results by age, ethnicity and parity is not discussed.
First of all women enrolled in the study were randomized according to age (18±29, 30±34 and >35),, parity (1 or >1) , ethnicity (Caucasian, hispanic and other) , and BMI. As the sample retained during follow-up at 2 years of age represents 80% of the mothers analyzed after delivery, we consider that it is appropriate to adjust the data of the descendants for these variables. It is also known that they can affect the health of descendants. According to your comments, we have added a detailed phrase and four new references (26-29) in page 9 line 194-195 to clarify these aspects. Advanced maternal age, parity and ethnicity are associated with worse health in the offspring. This is why results have been adjusted according to these variables.
I am methodologically concerned with the first limitation mentioned by the authors “differences in diet between groups may have not been wide enough to induce changes in offspring health”. Since the intervention in clinical trials must be totally differentiated from the recommendations given to the control group. Furthermore, part of the second limitation could have been resolved by showing results adjusted for more potentially confusing variables.
Indeed, the dietary recommendations were different between groups, which is reflected by both the Nutrition and MEDAS scores (shown in Table 1). In spite of this, we think that maybe these dietary differences were not remarkable enough as to generate significant differences in the impact during foetal development and, hence, in the children’s health at 2 years of age.
With regards to the second paragraph of the limitations section, you are right. However, our sample size is not high enough as to make adjustments for all the potentially confounding factors. For this reason, we have only adjusted the results by age, parity and ethnicity. As explained earlier, these factors have been associated with the offspring’s health. We still think that breastfeeding and the vaccinations could have influenced our results and therefore chose to include it in this section.
Figures. Figure 1 does not have the title in the text. The image quality is not good and cannot be appreciated well.
We agree with your comments. The quality of the image has been improved. We have also added a detailed title of the figure in the text (page 5, line 110).
Tables. It would avoid shading because it is very difficult to understand.
We agree your suggestion. The shading has been removed from the tables (pages 9-12, tables 1 and 2).
Reviewer 3 Report
In the manuscript authors assessed the effect of Mediterranean diet supplemented with extra virgin olive oil and pistachios during pregnancy on the health of children at the age of 2. The study is properly designed and meticulously described and surely will be of interest to all health professionals providing care to pregnant women and pediatricians. Conclusions are fully supported by the obtained results and limitations of the study are discussed. The language is good.
There are, however, certain implementations to be made before publication.
Major:
- The Cesarean section is a known risk-factor for developing certain autoimmune diseases in children. The information about CS rate in the studied population is lacking. Please provide data and discuss this matter.
Minor:
- In Table 2 13.6% of children were admitted to the hospital whereas in the text it was 13.9%. Please correct.
Author Response
Thank you very much for your kind comments and constructive suggestions. We agree with you on most points. We have made the corresponding changes to improve the article.
The changes applied are as follows:
In the manuscript authors assessed the effect of Mediterranean diet supplemented with extra virgin olive oil and pistachios during pregnancy on the health of children at the age of 2. The study is properly designed and meticulously described and surely will be of interest to all health professionals providing care to pregnant women and paediatricians. Conclusions are fully supported by the obtained results and limitations of the study are discussed. The language is good.
There are, however, certain implementations to be made before publication.
Major:
- The Cesarean section is a known risk-factor for developing certain autoimmune diseases in children. The information about CS rate in the studied population is lacking. Please provide data and discuss this matter.
We agree with you on the importance of adding this data. We have just added this information in the table (page 9-10, table 1) and discussed them (page 15, lines 271-274). The results about caesarean section in both groups are similar.
Minor:
- In Table 2 13.6% of children were admitted to the hospital whereas in the text it was 13.9%. Please correct.
We agree with your appreciation. We have just corrected the number in the text (page 11, line 218).
Round 2
Reviewer 1 Report
The Authors answered to all criticism.
However, I suggest inserting the parameter regarding SGA/LGA, etc in the results and discussing also them. The data are few and did not influence the results; however, an effect on a wider cohort cannot be excluded.
Author Response
cv
Thank you very much for your constructive comments.
Following your recommendations, we have inserted in table 2 the SGA, LGA and Preterm data. These are the only data which are different between CG vs IG (SGA, LGA). However, these cases are small to influence our results. This reason is added what was already commented in the responses of version 1: pharmacological treatments and events are only considered from the first visit to the paediatrician after hospital discharge (approximately 1 month later). Other rates in which there are no differences between groups or there are only few cases, have not been included in the table so as not to add non-relevant information.
We have also added this comment in the limitations:
For instance, the rate of SGA and LGA newborns. Although it is significantly lower in the IG, the number is small enough to affect our results. However, its effect in a wider cohort or an older age can not be ruled out.
Reviewer 2 Report
This article, after being reviewed by the authors, has improved the quality of the presentation.
Author Response
Thank you very much for your kind comments